# A Novel Localization Method of Wireless Covert Communication Entity for Post-Steganalysis

**Guo Wei** [1,2], **Shichang Ding** [1,2,*], **Haifeng Yang** [1,2], **Wenyan Liu** [1,2], **Meijuan Yin** [1,2] and **Lingling Li** [3]

1   State Key Laboratory of Mathematical Engineering and Advanced Computing, Zhengzhou 450001, China
2   Henan Province Key Laboratory of Cyberspace Situation Awareness, Zhengzhou 450001, China
3   School of Intelligent Engineering, Zhengzhou University of Aeronautics, Zhengzhou 450046, China
*   Correspondence: scdingwork@outlook.com

**Abstract:** Recently, some criminals have begun to use multimedia steganography to conduct covert communication, such as transmitting stolen trade secrets. After using steganalysis to find covert communication entities, obtaining their locations can effectively help criminal forensics. This paper proposes a novel localization method of wireless covert communication entity for post-steganalysis. The method is based on hybrid particle swarm optimization and gray wolf optimization to improve localization precision (ILP-PSOGWO). In this method, firstly, the relationship model between received signal strength (RSS) and distance is constructed for the indoor environment where the target node exists. Secondly, dichotomy is used to narrow the region where the target node is located. Then, the weighted distance strategy is used to select the reference point locations with strong and stable RSS. Finally, the intersection region of the reference points is taken as the region where the target node is located, and the hybrid PSOGWO is used to locate and optimize the target node location. Experimental results demonstrate that ILP-PSOGWO can maintain high stability, and 90% of the localization errors are lower than 0.9012 m. In addition, compared with the existing methods of PSO, GWO and extended weighted centroid localization (EWCL), the average localization error of ILP-PSOGWO is also reduced by 28.2–49.0%.

**Keywords:** multimedia steganography; wireless sensor node localization; post-steganalysis; RSS; PSOGWO

## 1. Introduction

With the continuous development of communication and network technology, multimedia steganography has been widely used in the fields of covert communication [1] and digital watermarking [2]. Recently, more and more criminals have begun to use multimedia steganography to commit high-tech crimes [3–5]. They hide secret information in public media information to make people hardly able to detect its existence with intuitive vision and hearing. Therefore, these criminals can achieve the purpose of transmitting secret information.

Although the existing steganography detection methods can successfully detect covert communication entities, the physical locations of the covert communication entities are still unknown. To realize the complete forensics of the crime, post-steganalysis that investigates the physical location of the covert communication entity should be carried out [6]. Therefore, locating the covert communication entities has important practical significance and research significance for national network information security and social security and stability [7].

Previous works have discussed how to locate the covert communication entities based on IP addresses [6], but IP geolocation is usually suitable for outdoor localization and cannot be used for indoor localization. How to locate the indoor locations of the entities relying on wireless sensor nodes for covert communication is still seldom discussed. Currently, illegal elements can easily hide illegal information in the data transmitted by wireless sensors to achieve malicious attacks [8]. To obtain the wireless sensor node location to help criminal

evidence, we can locate the wireless sensor node by studying its received signal strength (RSS) [9,10].

At present, the localization method based on RSS is widely used due to its low energy consumption, low cost and low implementation complexity [11–13]. Although the localization-method-based RSS is easy to implement, the localization precision is easily affected by environmental factors, such as multipath effect, non-line of sight and shadow fading. Therefore, some scholars have improved this method [14,15]. Kang et al. [14] propose an RSS self-calibration protocol to improve localization precision. An RSS self-calibration protocol can adopt calibration parameters according to the change in environment over time. Nguyen et al. [15] first used a deep learning method to refine the collected raw RSS data. Then, they adopted a machine learning method to extract the inherent spatial network geometric relationships from RSS data sets. These correlations are used to reduce the amplitude of RSS fluctuations and thus improve localization precision. In addition, some other localization methods can also be extended by methods based on RSS [16–21]. Zhao et al. [16] and Tang et al. [17] evaluate the distance between the reference point and the target node according to the obtained RSS value. They identify the overlapping region centroid of all reference points within the communication radius of the target node as the target node location. Finally, they propose the centroid localization method. Cho et al. [18] and Luo et al. [19] propose a localization method based on location fingerprint. This method uses RSS to construct a location fingerprint database, and finally realize localization. Han et al. [20] and Sun et al. [21] propose a method based on RSS approximation, which is based on the gradient value of the RSS. This method first measures the RSS of multiple points and then compares the RSS. Finally, it will analyze the changing trend of the RSS and estimate the target node location.

However, the existing RSS-based localization methods still have some problems in reference point selection and improving localization precision. For example, the WiMAP [22] and DWiMAP [23] mainly focus on reducing the selection of unstable reference points, but they ignore narrowing sampling regions, which increases the sampling workload and leads to the redundancy of reference point selection. The localization methods based on PSO [24] and GWO [25] are slow in convergence speed and easy to fall into local optimization, resulting in low localization precision. To address the above problems, this paper proposes a novel localization method of wireless covert communication entity for post-steganalysis, which is based on hybrid particle swarm optimization and gray wolf optimization to improve localization precision (ILP-PSOGWO). In this method, the relationship model between RSS and distance is firstly constructed for the indoor environment where the target node is located. Secondly, dichotomy is used to narrow the region where the target node is located. Then, the weighted distance strategy is used to select the reference point locations with strong and stable RSS. Finally, the intersection region of reference points is taken as the target node region, and then the hybrid PSOGWO is used to optimize the target node location. The main contributions of this paper are as follows:

- We propose to use the Gaussian filter model and the weighted distance strategy to select the reference point location. These two strategies can effectively reduce the influence of weak RSS and unstable reference points on the localization results and improve the localization precision.
- Through multiple measurements, we find that the measured RSS at the same location will fluctuate up and down within a certain range. Compared with the circle used in the traditional method, this paper uses the ring as the region where the target node exists. Therefore, it will effectively narrow the region where the target node may appear, and then improve localization precision and the stability of localization results.
- The hybrid PSOGWO algorithm is first used in the field of wireless sensor node localization. The algorithm not only improves the localization precision of target node but also promotes localization efficiency.

The rest of this paper is organized as follows. In Section 2, the related work and the basic principle of the hybrid PSOGWO algorithm are discussed. Section 3 describes

the basic principle of ILP-PSOGWO and gives the basic idea, main steps and complexity analysis of the algorithm. Section 4 verifies the performance of ILP-PSOGWO by comparing it with existing methods. Section 5 summarizes the paper and discusses the direction of further research.

## 2. Related Work

This section describes the existing relevant research on the localization method based on RSS and the hybrid PSOGWO used in this paper.

### 2.1. Localization Method Based on RSS

Ketkhaw et al. [26] propose a deep convolutional network localization method. This method firstly divides the localization region into multiple sub-regions, then places the target node in sub-regions (1,1) and measures the RSS from sub-regions (1,1) to (N, N); then, it moves to sub-region (1, 2) and repeats the same operation until moving to (N, N); finally, the deep neural network method is used to train and learn the timestamp, RSSI, SSID and MAC address information of the captured beacon frame to achieve locating. The implementation process of this method needs a lot of data collection, and the data need to be collected again once the environment changes. Awad et al. [22] put forward the WiMAP localization method, which divides the localization process of the target node into three stages: sample collection (select reference point), sample filtering and location evaluation. When collecting samples, the target region is sampled at equal intervals, then the acquired RSS is filtered, and finally, the geometric mean value is calculated only for the selected x% sample points. This method has the problems of redundant sampling and long sampling time. To reduce the number and time of sampling, Awad et al. [23] propose the DWiMAP method, which assigns different sampling densities according to the distance from the target node. Booranawong et al. [27] further improve the weighted centroid localization (WCL) method [28,29] and propose an extended weighted centroid localization (EWCL) method, which utilizes four reference points to locate. However, only three are deployed in the localization region. The distance information of these three reference points is used to estimate the distance between the target and the fourth reference point. This method is feasible in the theoretical study, but there may be accumulated errors in the actual test environment.

Alhammadi et al. [30] and Awad et al. [24] propose the particle swarm optimization (PSO) method to locate the target node. The main idea of [30] is that when the distance from the particle to all samples is equal to the distance from the target node to all samples, the particle location is selected as the target node location. However, the disadvantage of this method is that when the number of samples is large, the particles that can satisfy the condition are not unique, which will lead to high localization error. Awad et al. [24] propose a method based on RSS and PSO to locate the target node, which achieves localization by using multiple samples of RSS at known locations as the input of PSO algorithm. Although this method is easy to implement, it does not solve the problem that PSO is prone to fall into local optimum, resulting in slow convergence speed and low localization precision. To solve this problem, Cai et al. [31] improve the particle velocity but still do not solve the problem that particles are prone to fall into local optima.

In a word, the localization method based on RSS still has some problems in the selection of reference point location and locating the target node. Therefore, this paper proposes a novel localization method, which not only improves the selection strategy of reference points location, but also introduces a hybrid PSOGWO algorithm when locating the target node to achieve high precision and efficiency.

### 2.2. Hybird PSOGWO Principle

The hybrid PSOGWO algorithm has the characteristics of strong convergence ability and not easily falling into local optimum, so it has a broad application prospect [32–34].

Therefore, this section elaborates the basic principles of three meta-heuristic algorithms for forming hybrid ILP-PSOGWO.

### 2.2.1. PSO

Particle swarm optimization (PSO) [35,36] was first proposed in 1995. It is a population intelligent optimization algorithm inspired by the foraging behavior of birds. In PSO, the solution of each optimization problem represents a bird of the search space, called 'particle'. All particles update their speed and location by tracking two 'extremum'. About these two extrema, one is the optimal solution found by the particle itself, called the individual extreme value. The other is found in the current iteration of the whole population, called the global extremum. The updating equation of particle velocity and location during iteration is as follows:

$$V_{ij}^{t+1} = wV_{ij}^t + c_1 r_1 (Pbest_{ij}^t - L_{ij}^t) + c_2 r_2 (Gbest_j^t - L_{ij}^t) \qquad (1)$$

$$L_{ij}^{t+1} = L_{ij}^t + V_{ij}^{t+1} \qquad (2)$$

where $w$ is the inertial factor. By adjusting the size of $w$, the performance of global optimization and local optimization can be promoted. $c_1$ and $c_2$ are the acceleration constants. $c_1$ and $c_2$ represent the individual learning factor and social learning factor of each particle, respectively. $r_1$ and $r_2$ are random numbers in the interval [0,1], which can increase the randomness of the search. $Pbest_{ij}^t$ and $Gbest_j^t$ represent the historical optimal location of particle $i$ population in the $j$th dimension and $t$th iteration, respectively. $V_{ij}^t$ and $L_{ij}^t$ represent the velocity vector and location of particle $i$ in the $j$th dimension and $i$th iteration. The implementation process of PSO is shown in Algorithm 1, where the input is the velocity $V_i$ and the location $L_i$ of each particle and the maximum number of iterations $T$, and the output is the optimal solution $L_t$.

---

**Algorithm 1:** The implementation process of PSO.

**Input:** the location $L_i(i = 1, 2, \cdots, n)$ and velocity $V_i(i = 1, 2, \cdots, n)$ of each
      particle
**Output:** the optimal solution $L_t$

1   initialize $w$, $c_1$ and $c_2$
2   set the number of iteration $T$
3   set global best fitness as $g\_best$ and local best fitness as $p\_best$
4   **while** *(t < T)* **do**
5      **for** *each particle i* **do**
6         calculate the fitness value $f_i$ of each particle
7      **for** *each particle j* **do**
8         **if** $f_{ij} < p\_best$ **then**
9            $p\_best = f_{ij}, L\_pbest = L_i$
10     **if** $p\_best < g\_best$ **then**
11        $g\_best = p\_best, L\_gbest = L\_pbest$
12     update the location and velocity by Equations (1)–(2)
13     $t = t + 1$

---

### 2.2.2. GWO

Gray wolf optimization (GWO) is a swarm intelligence optimization search method developed by Seyedali et al. [25] inspired by the prey activity of gray wolves. GWO has been widely used by some scholars due to its strong convergence, fewer parameters and low implementation complexity and has been successfully applied to various fields [37,38]. The optimization process of GWO includes five stages of gray wolf social classification,

surrounding prey, searching prey, attacking prey and hunting. The specific content is described as follows:

**Stage 1: Divide the social hierarchy of gray wolves.** Calculate the fitness of each individual in the population, and the three gray wolves with the best fitness are labeled as $\alpha$, $\beta$ and $\delta$, and the remaining gray wolves are labeled as $\omega$. The optimization process is mainly guided by the three optimal solutions in each iteration of the population.

**Stage 2: Surround the prey.** When searching for prey, gray wolves gradually approach and surround it. The mathematical model of this behavior is as follows:

$$D = |C \times L_p(t) - L(t)|, C = 2r_2 \tag{3}$$

$$L(t+1) = L_p(t) - A \times D, A = 2a \times r_1 \tag{4}$$

where $t$ is the current number of iterations, $A$ and $C$ are the coordination coefficient vectors and $L_p$ and $L_t$ are the location vector of the prey and the gray wolf. $a$ linearly decreases from 2 to 1 during the entire iteration process. $r_1$ and $r_2$ are random vectors in [0, 1].

**Stage 3: Search for prey.** In each iteration, these three gray wolves ($\alpha$, $\beta$ and $\delta$) have a strong ability to identify the potential location of prey, and then the locations of other search agents (including $\omega$) are updated according to the best three gray wolves. The mathematical model of this behavior can be expressed as follows:

$$D_\alpha = |C_1 \times L_\alpha - L|, D_\beta = |C_2 \times L_\beta - L|, D_\delta = |C_3 \times L_\delta - L| \tag{5}$$

$$L_1 = |L_\alpha - A_1 \times D_\alpha|, L_2 = |L_\beta - A_2 \times D_\beta|, L_3 = |L_\delta - A_3 \times D_\delta| \tag{6}$$

$$L(t+1) = \frac{L_1 + L_2 + L_3}{3} \tag{7}$$

where $L_\alpha$, $L_\beta$ and $L_\delta$ represent the location vectors of $\alpha$, $\beta$ and $\delta$, respectively. $L$ represents the location vector of the prey. $D_\alpha$, $D_\beta$ and $D_\delta$ represent the distances between the current candidate gray wolf and the optimal three wolves, respectively.

**Stage 4: Attack the prey.** When the value of $a$ linear declines from 2 to 0, the corresponding value of $A$ also changes in the interval. When $|A| > 1$, gray wolves try to disperse in each region and search for prey. When $|A| < 1$, the gray wolves will concentrate their search and begin to attack the prey.

**Stage 5: Hunting.** In this stage, the optimal solution is obtained iteratively. The vector $C$ of the cooperation coefficient is a random value, which is conducive to the algorithm jumping out of the local, especially in the later stage of the iteration. The implementation process of GWO is shown in Algorithm 2; the input is the gray wolf population $n$ and the maximum number of iterations $T$, and the output is the optimal solution $L_t$.

---

**Algorithm 2:** The implementation process of GWO.

**Input:** gray wolf population $L_i (i = 1, 2, \cdots, n)$, size $n$, number of iteration $T$,
        control parameter $a$, synergy coefficient $A$, $C$ and learning factors $r_1$, $r_2$

**Output:** the optimal solution $L_t$

1　initialize the first best solution as $L_\alpha$, the second as $L_\beta$ and the third as $L_\delta$
2　**while** *(t < T)* **do**
3　　**for** $i = 1 : n$ **do**
4　　　calculate the fitness of each search agent $Fit_i$
5　　　update the current search agent location by Equations (3)–(6)
6　　evaluate the fitness of $Fit_i$
7　　update the $a$, $A$ and $C$
8　　update the best solution $L_\alpha$, $L_\beta$ and $L_\delta$ by Equations (3)–(4)
9　　$t = t + 1$

---

### 2.2.3. PSOGWO

Although PSO and GWO have been widely used in many fields, there are still some problems in the optimization process. PSO has strong search ability, but it easily falls into local optimum. GWO has a strong ability to develop solution space and does not easily fall into local optima, but the lack of information sharing among gray wolves leads to poor search ability. The hybrid PSOGWO algorithm [39] can maximize the advantages and minimize the disadvantages of the PSO and GWO. Therefore, this paper uses it to locate the target node. To increase information sharing among individual gray wolves, when GWO calculates the final location, it does not simply take the average location of the three optimal wolves as the target location but introduces the speed equation of PSO, which can effectively improve the search ability of GWO. The improvements of this hybrid model are as follows:

$$V_{ij}^{t+1} = wV_{ij}^t + c_1 r_1 (L_1 - L_{ij}^t) + c_2 r_2 (L_2 - L_{ij}^t) + c_3 r_3 (L_3 - L_{ij}^t) \tag{8}$$

$$L_{ij}^{t+1} = L_{ij}^t + V_{ij}^{t+1} \tag{9}$$

where $w$ is the inertial factor. By adjusting the size of $w$, the performance of global optimization and local optimization can be promoted. $c_1$, $c_2$ and $c_3$ are the acceleration constants and represent the learning factor of the three optimal gray wolves, respectively. $r_1$, $r_2$ and $r_3$ are random numbers in the interval [0, 1], which can increase the randomness of the search. $L_1$, $L_2$ and $L_3$ represent the locations of the optimal three gray wolves, respectively. $L_{ij}^t$ and $V_{ij}^t$ are the velocity vector and location of the candidate gray wolf $i$ (i.e., the optimal solution) in the $j$th dimension and $t$th iteration.

The implementation process of ILP-PSOGWO is shown in Algorithm 3.

---

**Algorithm 3:** The implementation process of PSOGWO.

**Input:** gray wolf population $L_i (i = 1, 2, \cdots, n)$, size $n$, number of iteration $T$, control parameter $a$, synergy coefficient $A$, $C$ and learning factors $r_1$, $r_2$

**Output:** the optimal solution $L_t$

1   calculate the fitness value $fit_i$ of each search agent
2   Initialize the first best solution as $L_\alpha$, the second as $L_\beta$ and the third as $L_\delta$
3   **while** *(t < T)* **do**
4      **for** $i = 1 : n$ **do**
5          calculate the fitness value $fit_i$ of each particle
6          update the fitness value of $\alpha$, $\beta$ and $\delta$, update $a$
7      **for** $i = 1 : n$ **do**
8          **for** $j = 1 : d$ **do**
9              calculate $D_\alpha$, $D_\beta$ and $D_\delta$ by Equation (5)
10             update the $r_1$, $r_2$, $A$ and $C$
11             calculate velocity $V_i$ and the location $L_t$ by Equations (6)–(9)

---

## 3. ILP-PSOGWO Method

The process of ILP-PSOGWO can be roughly divided into the following three stages: the construction and analysis of the relationship model between the RSS and distance, the selection of the reference point locations and locating the target node. The principle framework of ILP-PSOGWO is shown in Figure 1, and its detailed process is described as follows:

- **Step 1: Collect data.** Determine the room where the target node is located and start to collect the signal strength to provide data support for building the signal attenuation model.
- **Step 2: Process data.** To obtain more reliable data, we use Gaussian filtering to filter the signal strength obtained in Step 1.

- **Step 3: Fit Data.** The obtained signal strength and distance are fitted, and the relationship model between signal strength and distance is constructed.
- **Step 4: Evaluate model.** Use the model evaluation criteria to evaluate the relationship model between signal strength and distance constructed in step 3, and select a model with the best fitting degree.
- **Step 5: Determine and narrow the selected region of reference point.** Use the dichotomy to divide the room to be located, and further narrow the region where the target node is located.
- **Step 6: Deploy the location of reference points.** Several possible situations of reference point deployments are given, and the location of reference point deployment is limited.
- **Step 7: Adopt the weighted distance strategy to select the reference points.** The selection of reference points will directly affect the localization precision of the target node. This paper proposes to use the weighted distance strategy to select reference points with strong and stable signal strength.
- **Step 8: Use the PSOGWO algorithm to locate.** According to the location information obtained above, the hybrid PSOGWO algorithm is used to locate and optimize the target node location.

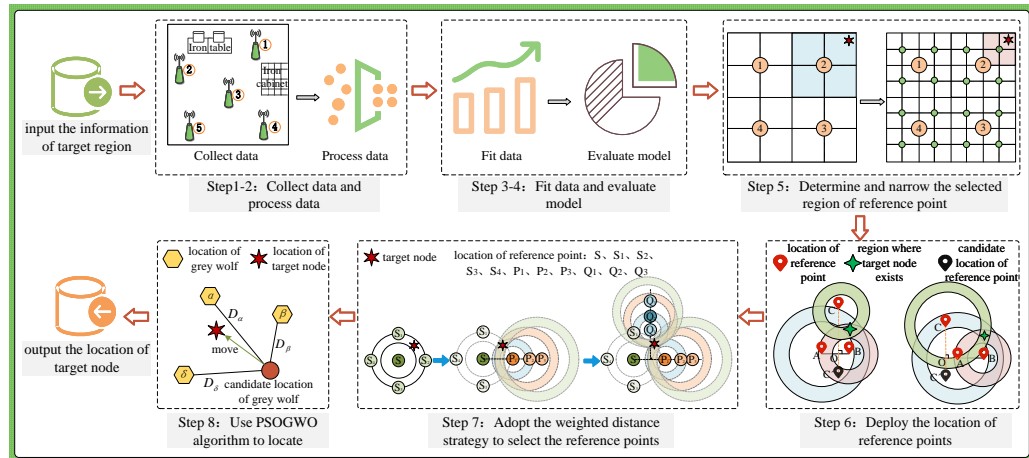

**Figure 1.** The basic principle framework of the ILP-PSOGWO.

The following chapters elaborate the above steps in detail.

### 3.1. Construction and Analysis of the Relationship Model between the RSS and Distance

This section is mainly divided into the following three stages: data acquisition and processing, RSS and distance relationship model construction and model evaluation. The following introduces the three stages, respectively.

### 3.1.1. Data Acquisition and Processing

The localization method based on RSS locates the target node by detecting the RSS of the target node. In this paper, the RSS is measured at different locations from the target node. As shown in Figure 2, the closer the distance to the target node, the stronger the RSS (i.e., the detection distance is inversely proportional to the RSS).

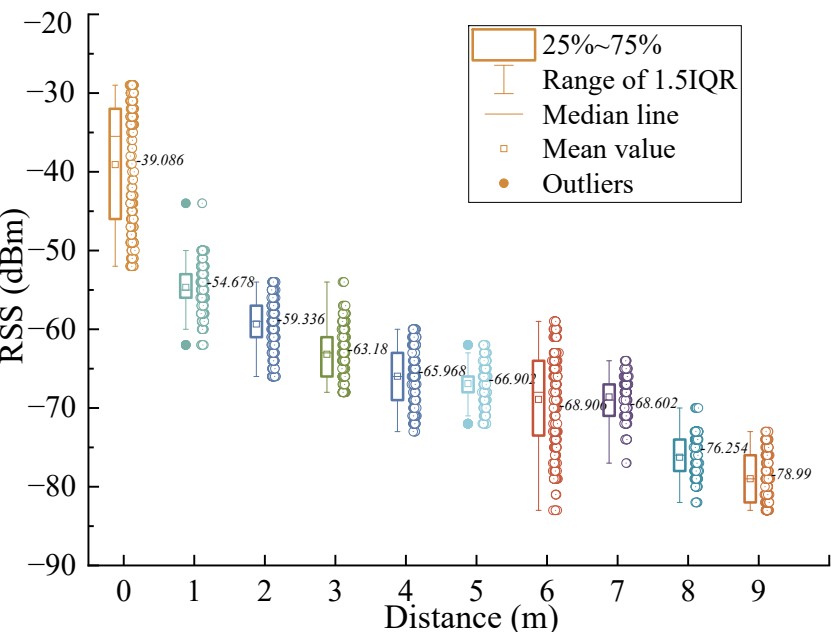

**Figure 2.** The relationship between distance and RSS.

Figure 2 also depicts that RSS is usually affected by reflection, refraction, noise, etc., in the transmission process, resulting in RSS fluctuating up and down. Therefore, before building the relationship model between RSS and distance, it is necessary to filter the obtained RSS.

Common filtering methods include mean filtering, median filtering, Gaussian filtering, etc. Considering that RSS follows or approximately follows the normal distribution, Gaussian filtering is selected to filter the obtained RSS [40]. In this method, the RSS value of the high-probability occurrence region is selected as the effective value by Gaussian model, and the average value of effective value is obtained as the measurement value. Suppose $n$ RSS values are collected from a reference point, which are $S_n = \{s_1, s_2, \cdots, s_n\}$, where the corresponding probability $P$ is $P_n = \{p_1, p_2, \cdots, p_n\}$ and mean and variance are $\mu$ and $\delta^2$, then the Gaussian filter model is

$$P(s) = \frac{1}{\delta\sqrt{2\pi}}e^{(-(-s-\mu)^2)/(2\delta^2)}, \tag{10}$$

where

$$\mu = \frac{s_1 + s_2 + \cdots + s_{n-1} + s_n}{n} = \sum_{i=1}^{n} s_i p_i \tag{11}$$

$$\delta^2 = \frac{1}{n-1}\sum_{i=1}^{n}(s_i - \mu)^2 \tag{12}$$

The minimum value of high probability is set to be greater than or equal to 0.6 to reduce the influence of small probability events on the collected RSS value. Therefore, when, by substituting the $\mu$ and $\delta$ obtained from Equations (11)–(12) into Equation (10), we can obtain the corresponding probability density of each RSS. After that, the filtered RSS can be obtained by sifting out the probability density less than 0.6, and we can use Equation (13) to obtain the optimized RSS.

$$RSS = \frac{1}{n}\sum_{i=1}^{n} x_i \tag{13}$$

Figure 3 shows the processing results of the RSS value. Gaussian filtering eliminates the deviation of the RSS value caused by noise to a certain extent, and the stability is better.

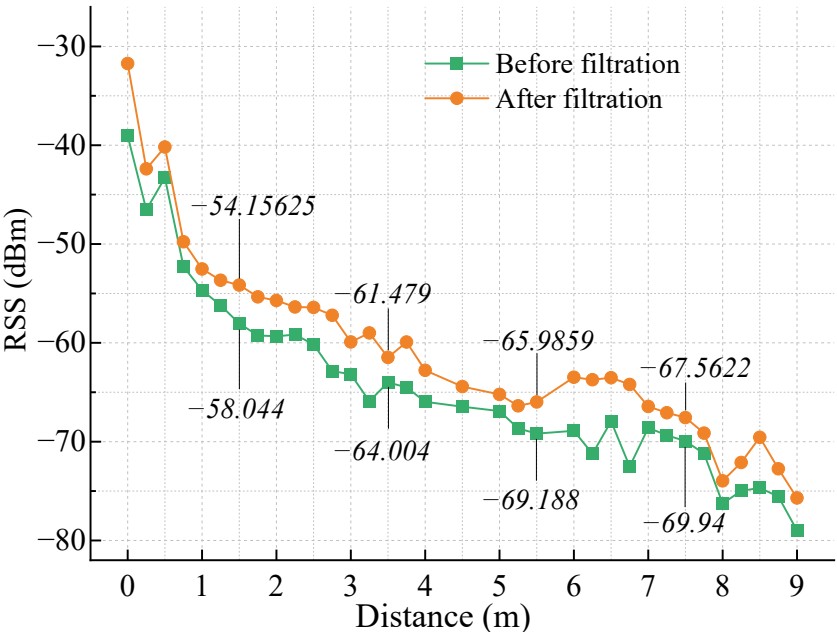

**Figure 3.** The RSS processing.

As shown in Figure 4, the distance within 0–9m is selected to measure the RSS obtained by the receiver, where the black curve represents the original data and the other curves are the fitted RSS attenuation model. In Figure 4, the other values represent the three sets of parameters corresponding to the three fitted models, Logistic, Log3P1 and Poly4. Each set of parameters can ensure that the corresponding model achieves an optimal fit. After comparison with a variety of curve models, three kinds of models with good fitting effect, Logistic, Poly4 and Log3P1, are selected to construct the relationship model between RSS and distance.

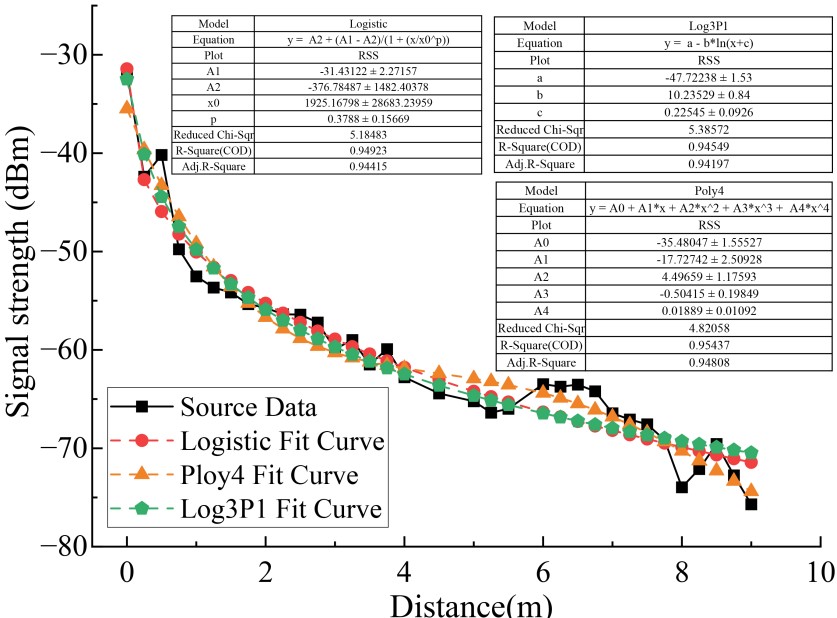

**Figure 4.** The RSS processing.

### 3.1.2. Model Evaluation

To further select a signal attenuation model with the best fitting effect, this paper evaluates the three types of models from the following three criteria:

Reduced Chi-Sqr: The sum of the squared variances of the estimated and actual $y$ values at each point is called the sum of residual squares. The smaller the sum of residual squares, the better the fitting effect of the model. Figure 4 shows that the minimum residual sum of squares of the Poly4 model is 4.82058, and the maximum residual sum of squares of the Log3P1 model is 5.38572.

R-Square($R^2$): The closer $R^2$ is to 1, the better the fitting result is. Among the three models, the $R^2$ of log3p1 model is 0.95437, and the fitting effect is the best.

Adjusted $R^2$: The larger the gap between adjusted $R^2$ and $R^2$, the worse the fitting of the model. Figure 4 indicates the gap between $R^2$ and $R^2$ adjusted by logistic, and poly4 and log3p1 of the three models are 0.00508, 0.00629 and 0.00352, respectively. The gap between $R^2$ and $R^2$ adjusted by the three models is small, so the fitting effect is equivalent.

To sum up, this paper selects poly4 curve to build the relationship model between RSS and distance. The equation of the poly4 fitting curve is as follows:

$$y = 0.01889x^4 - 0.50415x^3 + 4.49659x^2 - 17.72742x - 35.48047 \tag{14}$$

### 3.2. Selection Strategy of Reference Point Location

The selection strategy of reference point location includes the following three stages: determine and reduce the selecting region of the reference point, adopt deployment strategy of reference point location and use the weighted distance strategy for selecting reference points. These three stages are introduced in detail below.

### 3.2.1. Determine and Narrow the Selected Region of Reference Point

After determining the target node is in a specific room or region, this paper adopts the dichotomy to binary the region. As shown in Figure 5a, we detect the RSS at the center of the divided four regions. If multiple regions are detected, we select the region with the largest RSS (i.e., the blue region in the figure) as the next sampling region. If the RSS is not detected, as shown in Figure 5b, we divide the four regions again, sample at the green point, and then select the next sampling region (i.e., the pink region in Figure 5b). Through multiple measurements, we found that the measured RSS at the same location will fluctuate up and down within a certain range. Compared with the circle used in the traditional method, this paper uses the ring as the region where the target node appears, as shown in Figure 6. Assuming that the range of RSS values measured at the reference point is $[-53, -45]$, it is converted into a distance $[d_2, d_1]$, that is, the ring formed by this distance range is the real region where the target node is located. Therefore, the RSS collected by a reference point will no longer be represented by a specific circle (i.e., the circle with the radius of $d$ in Figure 6) but the ring formed by the floating range of the RSS.

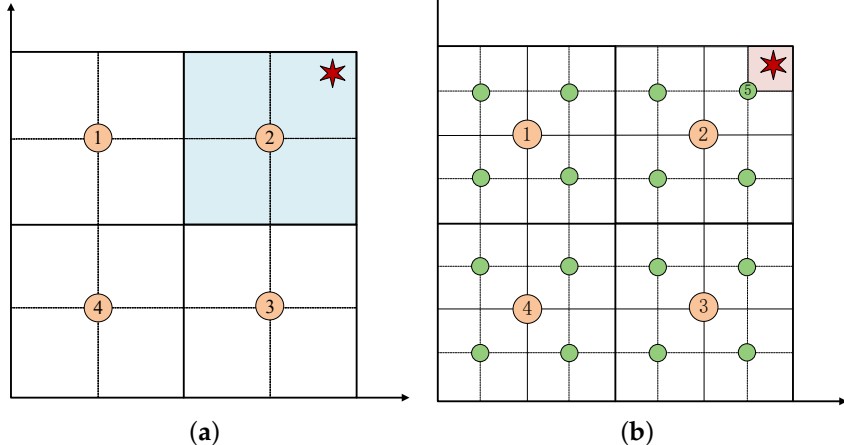

(a) (b)

**Figure 5.** Node distribution and regional division. (**a**) Case 1. (**b**) Case 2.

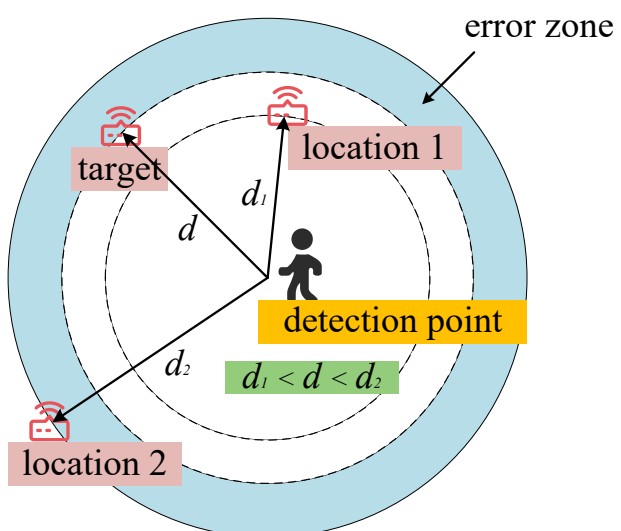

**Figure 6.** The region where the target node is.

### 3.2.2. Deployment Strategy of Reference Point Location

The dichotomy can effectively narrow the approximate region of the target node. After the approximate region is determined, this paper selects as few reference points as possible to locate the target node. As shown in Figure 7, when three reference points are collinear, there may be the following three situations: the first is that there is no intersection region, as shown in Figure 7a; the second is that there is a large intersection region, as shown in Figure 7b; and the third is that there are multiple intersection regions, as shown in Figure 7c. The above three cases increase the difficulty of locating the target node. To avoid the above situation, this paper limits the reference point location. When the number of reference points is equal to or greater than 3, we limit the selection of the third reference point location, so that the third reference point location is not on the extension line or reverse extension line between the first two reference point locations. In this way, we can realize that at least three reference point locations are non-collinear, and the concrete example is shown in Figure 8. As shown in Figure 8, we select the first two reference point locations $A$ and $B$. To avoid the collinearity of the three reference point locations, the selection of the third reference point location has the following three cases: the first is shown in Figure 8a, where the third reference point location $C$ or $C'$ is on the vertical line of the $AB$ line segment; the second case is shown in Figure 8b, where the location $C$ or $C'$ is on the perpendicular of the $AB$ extension; the third is shown in Figure 8c, where the location $C$ or $C'$ is on the perpendicular of the $AB$ reverse extension. At this time, it can be ensured that the ring formed by the three reference points has only a small intersection region.

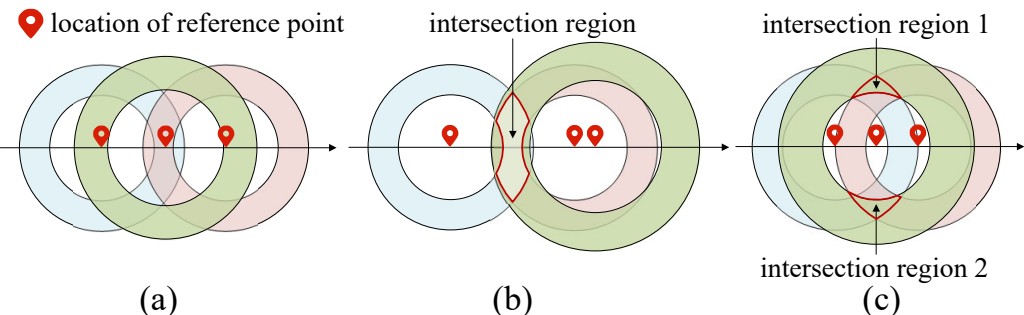

**Figure 7.** Collinear situation of reference points. (**a**) Case 1. (**b**) Case 2. (**c**) Case 3.

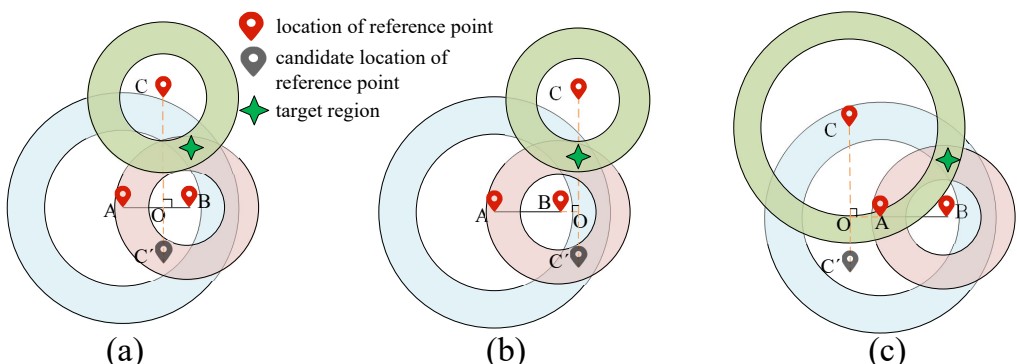

**Figure 8.** Deployment process of reference points. (**a**) Case 1. (**b**) Case 2. (**c**) Case 3.

### 3.2.3. Weighted Distance Strategy for Selecting Reference Points

The RSS and stability of the target node are different from the reference point location. As shown in Figure 9, this paper selects four different reference points $D_1$, $D_2$, $D_3$ and $D_4$. The RSS mean values of these four reference points are $-30.26$, $-50.24$, $-65.05$ and $-80.54$, respectively, and the variances are 1.952, 7.822, 6.868 and 3.028, respectively. Therefore, although the RSS at the reference point $D_2$ is strong, the signal stability is the worst. If the RSS is considered as the only criterion for selecting reference points, the localization precision may be reduced.

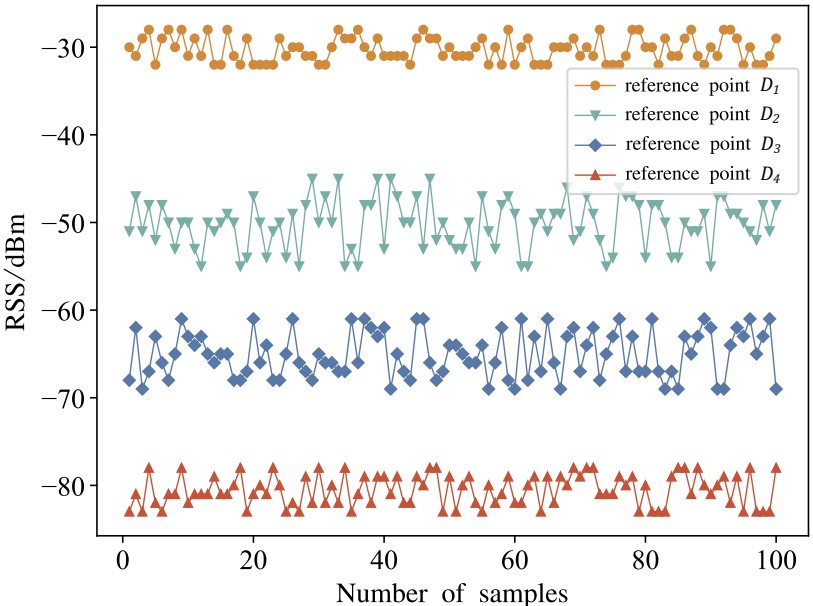

**Figure 9.** Collect the RSS.

Therefore, to select the reference points with strong and stable RSS, this paper records the variance of the RSS of the target node received by reference points to improve the reliability of reference point selection. According to the recorded variance, different weight coefficients are assigned to different reference points. The RSS collected continuously for $N$ times at the $i$th reference point can be expressed as follows:

$$RSS_i = (RSS_1^i, RSS_2^i, \cdots, RSS_n^i), n = 1, 2, \cdots, N. \tag{15}$$

According to the obtained $RSS_i$, we can calculate the variance $\sigma^2$ of the reference point. The larger the variance of the reference point is, the more unstable the RSS is and

the smaller the weight that should be allocated. The weight $w^i$ of the $i$th reference point is calculated as follows:

$$w^i = \frac{\frac{1}{\sigma^2+1}}{\sum_{i=1}^{t} \frac{1}{\sigma^2+1}} \tag{16}$$

where $t$ represents the number of reference points. The weighted distance $d_{ij}$ between the reference point and the target node after improvement can be calculated as follows:

$$d_{ij} = d_i \times w^i \tag{17}$$

where $j$ represents the target node, and $d_i$ represents the distance between the $i$th reference point and the target node. Taking the deployment of three reference points as an example, as shown in Figure 10a, $S$ is the first deployed reference point. To deploy the next reference point closest to the target node, we select a location in the upper, lower, left and right directions of location $S$, respectively, and compare the RSS of these four locations. Then, we select the location with the strongest RSS (i.e., $S_2$) as the candidate location for the second reference point. In Figure 10b, $P_1$, $P_2$ and $P_3$ represent three reference points selected within the range $[R_1, 2R_1]$ of distance $S$. According to the weighted distance strategy, if the weighted distance at $P_1$ is the shortest, this location is selected as the second reference point. Figure 10c shows the selection process of the third reference point. According to the deployment strategy of reference points, this paper selects the third reference points on the vertical bisector of location $S$ and $P_1$, namely $Q_1$, $Q_2$ and $Q_3$. The connecting distance between the third reference points and the first two reference points meets the range $[R_1, 2R_1]$. According to the weighted distance strategy, $Q_2$ will be selected as the third reference point. Finally, the third reference points, $S$, $P_1$ and $Q_2$ are selected in turn.

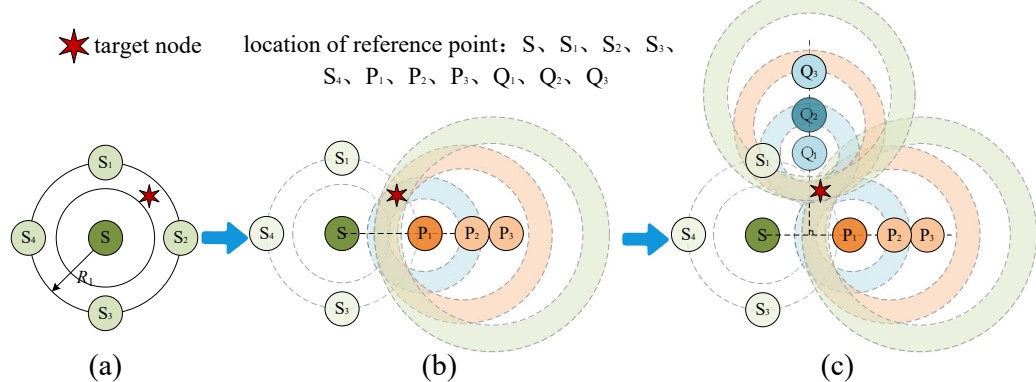

(a)                  (b)                  (c)

**Figure 10.** Select reference point. (**a**) Select the first reference point. (**b**) Select the second reference point. (**c**) Select the third reference point.

### 3.3. Locating the Target Node

As shown in Figure 11, simply taking the centroid of the intersection region of the three rings (i.e., the green hexagonal star) as the true location of the target node will cause a reduction in localization precision. Therefore, this paper uses hybrid PSOGWO to locate and optimize the target node location. The main processes of locating are as follows.

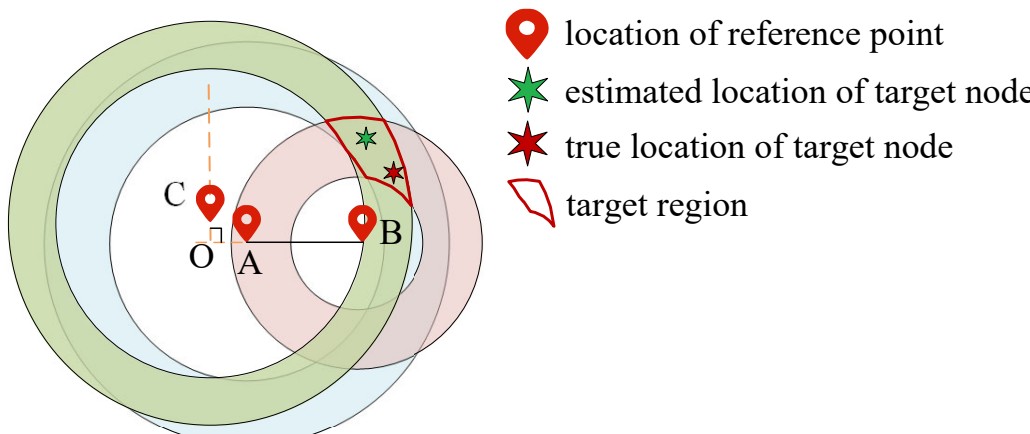

**Figure 11.** Locating target node.

- **Step 1:** As shown in Figure 12a, initialize the gray wolf population $N$ in the target region, and each gray wolf represents the potential location of the target node.

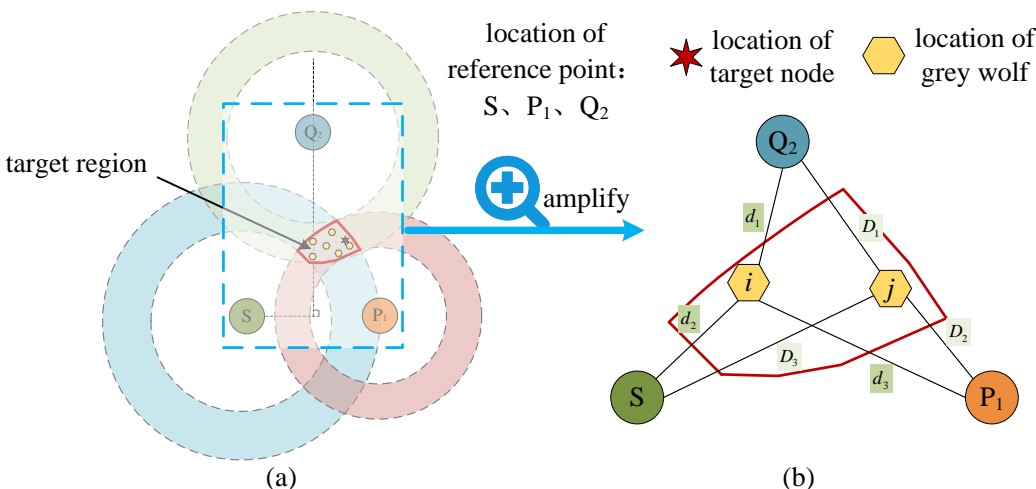

(a)                    (b)

**Figure 12.** Optimizing the target node location using PSOGWO. (**a**) Optimizing the target node location. (**b**) The situation of equal fitness values.

- **Step 2:** Calculate the fitness of gray wolf individuals and save the first three gray wolves with the best fitness $\alpha$, $\beta$ and $\delta$. The calculation equation of fitness value is as follows:

$$RMS_i = \sqrt{\frac{1}{n} \sum_{t=1}^{n} (d_{t,p} - d_{t,ap})^2}, \tag{18}$$

where

$$d_{t,p} = \sum_{t=1}^{n} \sqrt{(x_t - x_p)^2 + (y_t - y_p)^2}, n = 3 \tag{19}$$

$$d_{t,ap} = \sum_{t=1}^{n} \sqrt{(x_t - x_{ap})^2 + (y_t - y_{ap})^2}, n = 3 \tag{20}$$

where $RMS_i$ represents the fitness value of gray wolf $i$. The smaller the fitness value, the closer the actual location of the gray wolf and target node. $(x_t, y_t)$ and $(x_p, y_p)$ represent the reference point location and each gray wolf, $d_{t,p}$ is the distance from the reference point to each gray wolf, $d_{t,ap}$ is the distance from the reference point to the target node and $n$ is the number of reference points. The fitness ranking of the gray wolf may encounter the situation of equal fitness values, as shown in Figure 12b; the distance between gray wolf $i$ and $j$ to the three reference points are equal, that is,

$d_1 + d_2 + d_3 = D_1 + D_2 + D_3$. At this time, the fitness values of the two gray wolves are also equal, indicating that the gray wolf locations coincide, but in practice, gray wolf $i$'s location is not equal to gray wolf $j$'s location, which will affect the target node location. Therefore, if the fitness values of gray wolves are equal, this paper proposes to use the following conditions for exclusion:

$$d_{1,i} = d_{1,j}, d_{2,i} = d_{2,j}, \cdots, d_{n-1,i} = d_{n-1,j} \tag{21}$$

where $d_{1,i}$ and $d_{1,j}$ represent the distance from gray wolf $i$ and $j$ to the first reference point, respectively.

- **Step 3:** Update the current location of the gray wolf.
- **Step 4:** Update $a$, $A$ and $C$.
- **Step 5:** Calculate the fitness of all gray wolves.
- **Step 6:** Update the fitness and location of $\alpha$, $\beta$ and $\delta$.
- **Step 7:** Judge whether the maximum number of iterations or the fitness threshold is reached. If yes, exit the loop and output the target node location. Otherwise, go to Step 3 and continue.

The time complexity will affect the execution efficiency of the algorithm. The time complexity of ILP-PSOGWO is mainly composed of two aspects: the first is the dichotomy used to narrow the selected region of reference point, and the second is the hybrid PSOGWO used to optimize the target node location. The optimal time complexity of dichotomy is $O(1)$, and the worst time complexity is $O(\log_2 n)$. The time complexity of hybrid PSOGWO is $O(t \times n) + O(t \times n \times d)$. Therefore, the time complexity of the two methods is $O(\log_2 n)$ and $O(n^3)$, respectively. The time complexity of ILP-PSOGWO is $O(n^3)$.

## 4. Experiment

To verify the effectiveness and feasibility of ILP-PSOGWO, this paper obtains data sets in the actual environment, conducts experiments many times and compares the experimental results with the existing methods particle swarm optimization (PSO), gray wolf optimization (GWO) and extended weighted centroid localization (EWCL).

### 4.1. Experiment Setting

In real scenarios, the transmission environment of wireless communication systems is usually divided into line of sight (LOS) and not-line of sight (NLOS). In LOS, the radio signal can travel in a straight line between the transmitter and the receiver without any obstruction. If the conditions are not met, the RSS will be significantly attenuated (i.e., NLOS). To verify the impact of environmental complexity on ILP-PSOGWO, experiments are carried out in LOS and NLOS. The experiment was carried out on Intel(R) Core(TM) i7-11700@2.50 GHz with 32 GB RAM and Python 3.7.10 using Windows 10 Professional Edition platform. The detailed parameter settings in the experimental process are shown in Table 1.

**Table 1.** Experimental settings.

| Parameters | Value |
| --- | --- |
| The number of target nodes | 1 |
| Experimental environment | LOS, NLOS |
| The number of reference points | 3, 5, 7, 9, 11, 13 |
| Area of localization region (m$^2$) | 15, 20, 25, 30, 35, 40 |
| The number of iterations | 5, 10, 15, 20, 25, 30, 35 |
| The number of experiments | 5, 10, 15, 20, 25, 30, 35, 40, 45, 50, 100 |
| Test points location | (2.5, 5), (8, 7.3), (14.8, 6.2), (5.5, 9), (11.2, 7.8), (16, 5.1) |

### 4.2. Performance Evaluation

To verify the effectiveness and feasibility of ILP-PSOGWO, four aspects, including the LOS and NLOS environment, area of localization region, the number of reference points and number of iterations, are selected to evaluate the precision of the proposed method. Precision is the main index to evaluate the performance of a localization method, and it is also an important standard to evaluate the quality of a set of localization models. Localization precision is usually measured using the Euclidean distance between the estimated location and the true location, known as the localization error. The smaller the localization error, the higher the localization precision. The localization error is denoted by $error_i$, and it can be calculated as follows:

$$error_i = \sqrt{(x_{true} - x_{est})^2 + (y_{true} - y_{est})^2} \tag{22}$$

where $(x_{true}, y_{true})$, $(x_{est}, y_{est})$ represent the actual and estimated location of the target node $i$, respectively.

#### 4.2.1. Impact of Area of Localization Region on Localization Precision

When the number of reference points is 5, we tested the impact of different areas of the localization region on the localization precision of the four methods in indoor LOS and NLOS, respectively.

Figure 13a describes the influence of the area of localization region on the localization precision in LOS. With the increase in area, the localization error of ILP-PSOGWO and EWCL methods increases. The localization error obtained by PSO and GWO methods shows an unstable trend. This is because they search for feasible solutions in the whole localization region, and the search direction and speed are not fixed, leading to unstable localization results. ILP-PSOGWO narrows the search space of feasible solutions before using the PSOGWO, so it can obtain stable and more accurate localization results. Figure 13b indicates the influence of the area of localization region on the localization precision in NLOS. The localization errors of the four methods increase accordingly. Compared with EWCL and ILP-PSOGWO, the PSO and GWO are more sensitive to the environment and have larger localization errors.

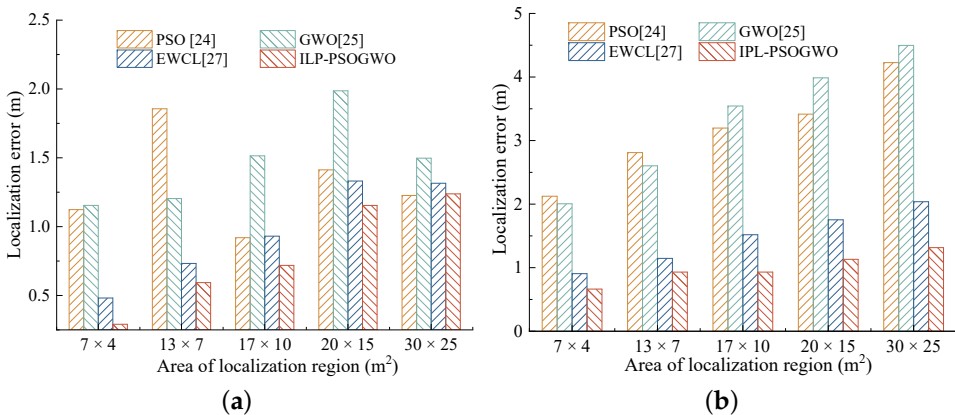

**Figure 13.** Impact of area of localization region on localization error. (**a**) LOS. (**b**) NLOS.

#### 4.2.2. Impact of Location of Test Point on Localization Precision

During the experiment, when the area of localization region is $17 \times 10$ and the number of reference points is 5, we assume that there are 6 target nodes needed to be located, and their locations are different. The experimental results are shown in Table 2. Compared with NLOS, the localization errors obtained by the four methods in the LOS are lower. This is because in the NLOS, the RSS is easily affected by the environment. Table 2 also describes that the localization errors measured by PSO and GWO in the two environments are quite different. However, the difference in localization errors measured by EWCL

and ILP-PSOGWO is little in the LOS and NLOS, indicating that the two methods have good stability.

**Table 2.** The impact of target node location on localization error.

| Environmental Environment | Localization Method | Test Points Location | | | | | |
|---|---|---|---|---|---|---|---|
| | | (2.5, 5) | (8, 7.3) | (14.8, 6.2) | (5.5, 9) | (11.2, 7.8) | (16, 5.1) |
| LOS | PSO [24] | 0.9196 | 1.3290 | 0.9832 | 1.2939 | 1.2037 | 0.9987 |
| | GWO [25] | 1.5143 | 1.4980 | 1.0372 | 1.3258 | 1.1835 | 0.8999 |
| | EWCL [27] | 0.8675 | 1.0903 | 0.8099 | 0.9099 | 0.7936 | 0.7091 |
| | ILP-PSOGWO | 0.7186 | 0.8021 | 0.5289 | 0.6503 | 0.5910 | 0.4521 |
| NLOS | PSO [24] | 2.0326 | 2.5710 | 1.7825 | 2.0627 | 2.0320 | 2.1833 |
| | GWO [25] | 2.7921 | 2.3467 | 1.9732 | 2.1362 | 2.2693 | 1.7902 |
| | EWCL [27] | 1.1290 | 1.2293 | 0.9021 | 1.1798 | 0.8998 | 0.9098 |
| | ILP-PSOGWO | 0.9027 | 1.0363 | 0.7936 | 0.9831 | 0.7352 | 0.5621 |

4.2.3. Impact of the Number of Reference Point on Localization Precision

The number of reference points is one of the factors that affect the localization precision of the target node. To observe the influence of the number of reference points on the localization precision, when the area of localization region is $17 \times 10$, we select the number of reference points as 3, 5, 7, 9, 11 and 13 for the experiment.

Figure 14 depicts the influence of the number of reference points on the localization precision of the three methods in LOS and NLOS. Figure 14a shows that when the number of reference points increases, the localization errors of the three methods decrease accordingly, and the localization errors of the three methods are all kept within 1.5 m. Figure 14b reveals that compared with GWO and ILP-PSOGWO, the change in the number of reference points has a greater impact on the localization results of PSO. This is because although the PSO has a strong ability to develop new space, it lacks communication between particles and is prone to fall into local optima, resulting in unstable localization results. Although the GWO has a weak ability to develop new space, resulting in low localization precision, it can maintain high stability in localization results because of the information sharing among gray wolves. The ILP-PSOGWO significantly narrows the region where the target node exists, so it can maintain high localization precision when the number of reference point changes.

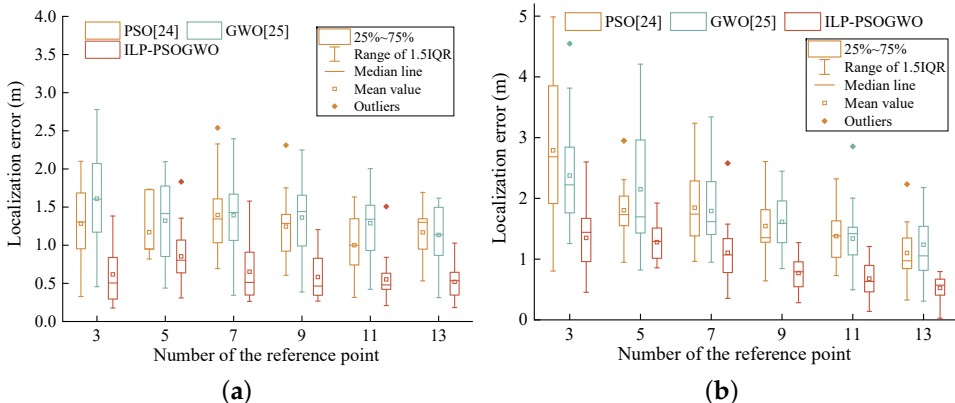

**Figure 14.** Impact of the number of reference point on localization error. (**a**) LOS. (**b**) NLOS.

4.2.4. Impact of Iteration Times on Localization Precision

Figure 15a describes the relationship between the number of iterations and the localization error in LOS. With the increase in iteration times, the localization errors of three methods are significantly reduced. When the number of iterations exceeds 15, the localization errors obtained by the three methods remain unchanged. The localization errors

of the PSO and GWO remain around 1 m, while the localization errors of ILP-PSOGWO are around 0.4 m. In addition, Figure 15a also indicates that ILP-PSOGWO achieves the optimal solution faster, which means it has strong convergence ability. This is because the hybrid PSOGWO combines the advantages of PSO and GWO, which makes it have a strong search ability and convergence ability. Figure 15b depicts the experimental results in NLOS. The environment changes lead to a significant increase in the localization errors of the three methods, which have a greater impact on the localization errors of PSO and GWO, and a smaller impact on the ILP-PSOGWO. Moreover, when the number of iterations increases to 35, the localization errors of PSO, GWO and ILP-PSOGWO are significantly reduced, which are 1.9073 m, 1.8351 m and 0.9112 m, respectively.

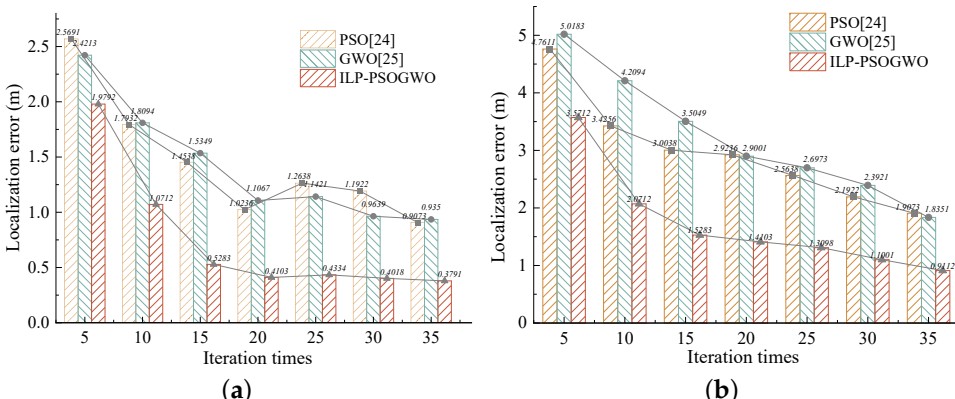

**Figure 15.** Impact of iteration times on localization error. (**a**) LOS. (**b**) NLOS.

### 4.2.5. Stability

To verify the stability of ILP-PSOGWO, the following two aspects are evaluated in this paper: standard deviation and cumulative probability distribution.

(i) Standard deviation

Standard deviation can reflect the degree of dispersion of a group of data, thus reflecting the stability of the group of data. The standard deviation $error_{std}$ can be calculated as follows:

$$error_{std} = \sqrt{\frac{\sum_{j}^{n}(error_j - error_{avg})^2}{n}} \tag{23}$$

where $n$ represents the number of experiment, $error_j$ represents the localization error of the $j$th measurement, which can be calculated in Equation (22), and $error_{avg}$ represents the average localization error of a group of data after the $j$th measurement.

When the number of sample points is 7 and the area of the localization region is $17 \times 10$, we conduct 5, 10, 15, 20, 25, 30, 35, 40, 45 and 50 experiments in LOS and NLOS, respectively. The experimental results are shown in Figure 16. In LOS, the standard deviation of PSO and GWO is below 0.7, and with the change in the number of experiments, the standard deviation also fluctuates greatly. EWCL and ILP-PSOGWO have good stability, and the standard deviation always stays below 0.4. In NLOS, the stability of the four methods is significantly decreased, but compared with EWCL and ILP-PSOGWO, PSO and GWO are less stable, and PSO even has the worst stability.

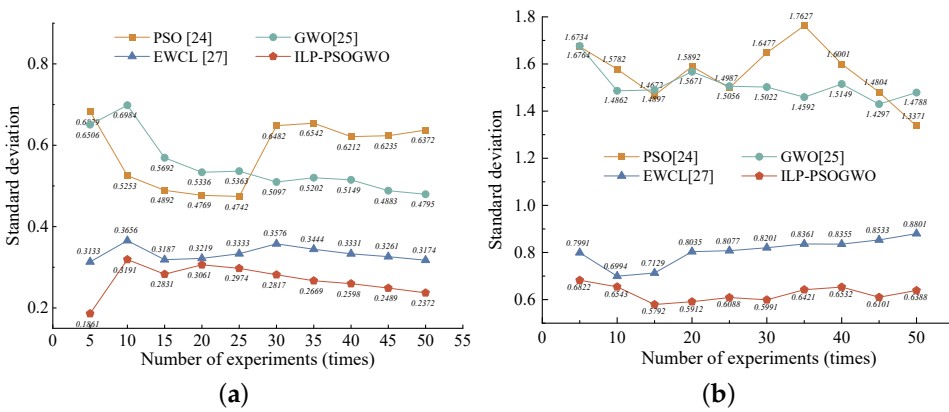

**Figure 16.** Standard deviation. (**a**) LOS. (**b**) NLOS.

(ii) Cumulative probability distribution

To further evaluate the stability of ILP-PSOGWO, this paper conducts 100 experiments on a target node location in LOS, and the experimental results are shown in Figure 17.

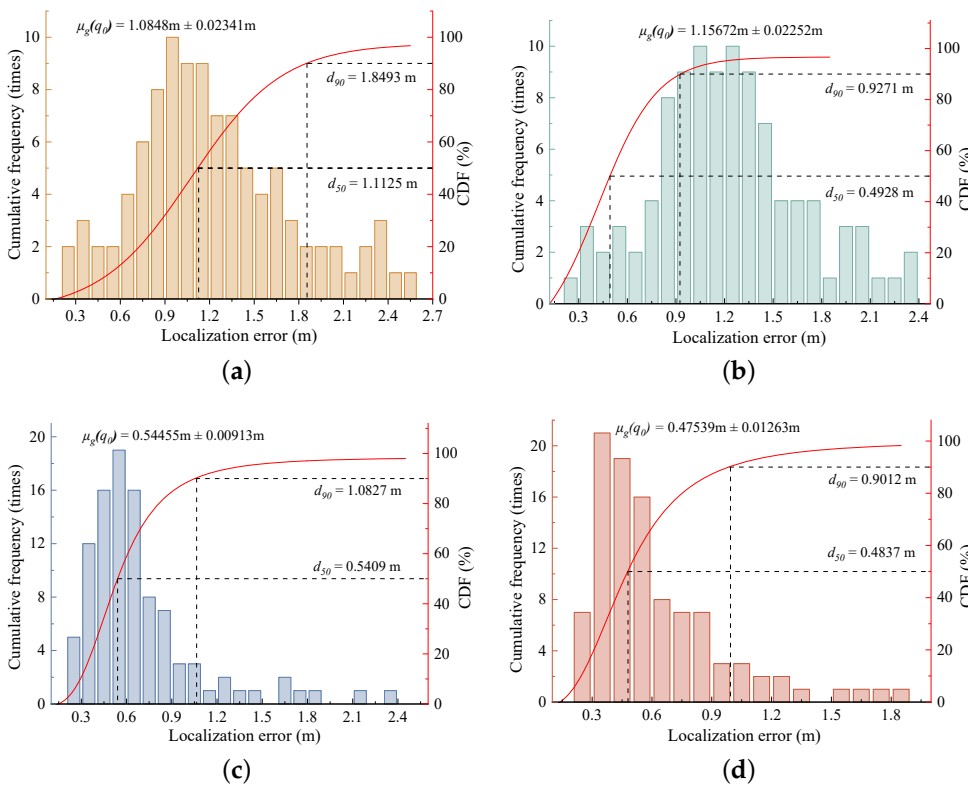

**Figure 17.** Cumulative probability distribution. (**a**) PSO. (**b**) GWO. (**c**) EWCL. (**d**) ILP-PSOGWO.

Figure 17 depicts the cumulative frequency distribution and cumulative frequency distribution of 100 localization errors obtained by the four methods. The average localization errors obtained by PSO, GWO, EWCL and ILP-PSOGWO are 1.0848 m, 1.15672 m, 0.54455 m and 0.47539 m, respectively. Figure 17 also shows that the localization errors obtained by PSO and GWO are mostly distributed between 0.6m and 1.8 m, and the frequency of occurrence is fewer than 10 times, indicating that the stability of these two methods is poor. In contrast, the localization errors of EWCL and ILP-PSOGWO are mostly distributed between 0.3 m and 0.7 m, and the frequency of occurrence is as high as 20, indicating good stability. Figure 17d indicates that 90% of the localization errors of ILP-PSOGWO are less than 0.9012 m, and 50% are less than 0.4837 m, which is better than the other three methods.

## 5. Conclusions

To obtain the indoor location of a wireless covert communication entity for post-steganalysis, this paper proposes a novel localization method based on hybrid particle swarm optimization and gray wolf optimization to improve the localization precision (ILP-PSOGWO). To verify the effectiveness and feasibility of ILP-PSOGWO, we carried out a series of comparative experiments. The experimental results indicate that ILP-PSOGWO can achieve higher localization precision and convergence speed. In addition, experimental results in the LOS and NLOS environments show that ILP-PSOGWO can maintain high stability in localization results. In future research, we will continue to focus and follow up on the indoor localization method of a wireless covert communication entity for post-steganalysis and further improve the localization precision of wireless covert communication entities.

**Author Contributions:** G.W. designed the study. G.W. and H.Y. performed the experiments and analyzed the data. G.W. wrote the paper. S.D. and W.L. reviewed the manuscript. M.Y. and L.L. guided the research. All authors have read and agreed to the published version of the manuscript.

**Funding:** This work was supported by the National Nature Science Foundation of China (No. 62172435, No. U1804263 and No. 61872449), Zhongyuan Science and Technology Innovation Leading Talent Project of China (No. 214200510019).

**Institutional Review Board Statement:** Not applicable.

**Informed Consent Statement:** Not applicable.

**Data Availability Statement:** The data presented in this study are available on request from the corresponding author.

**Conflicts of Interest:** The authors declare no conflict of interest.

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
