# Peer review of "A Novel Localization Method of Wireless Covert Communication Entity for Post-Steganalysis"

_applsci, doi:10.3390/app122312224_

Round 1

Reviewer 1 Report

Refer to the attached review report.

Author Response

Thank you for the comments concerning our manuscript. We have studied these comments carefully and have made corrections that we hope to meet with approval.

Reviewer 2 Report

- The authors must highlight the drawback or weak points for the previous work which will be addressed in this paper a bit more.

The paper provides a good discussion of all obtained results, and it is supported by simulation results.

Author Response

(The authors gave the same response as above.)

Reviewer 3 Report

In this paper “A Novel Localization Method of Wireless Covert Communication

Entity for Post-steganalysis”, authors propose a new novel localization method of wireless covert communication entity for post-steganalysis.

Authors start by detailing the algorithms used in this paper including its implementation.  After they describe the new method proposed. After they present the obtained results using real data. At the end of the manuscript, they present the conclusions of their work.

Although this manuscript can be relevant considering the actual state of the art, I can only recommend for publication after some minor clarifications.

Comments:

In my opinion, the stat of the art could be improved since we can found some works related with indoor localization in the context of wireless covert communications.

Besides that, it is referred by authors that “Recently, more and more criminals have begun to use multimedia steganography to commit high-tech crimes”. Nevertheless, the reference that it is presented is from 2010. Can authors clarify in the manuscript that?

Can authors please clarify in manuscript why the specific values (0.15 and 3.09) in equation (13). The same for Fig. (4).

Regarding Eq. (15), what represent those constants obtained from the fitting?

Finally, Figure 17 is quite confusing. Can authors simply that figure. The re is too much information.

Author Response

(The authors gave the same response as above.)
